# CSR and Long-Term Corporate Performance: The Moderating Effects of Government Subsidies and Peer Firm's CSR

Wenli Zhao [iD], Guangyu Ye, Guangyi Xu * [iD], Chong Liu *, Dandan Deng and Ming Huang

School of Business Administration, South China University of Technology, Guangzhou 510640, China; zhaowenlilong@163.com (W.Z.); bmgyye@scut.edu.cn (G.Y.); sabrina0771yaya@163.com (D.D.); huangming.0808@163.com (M.H.)
* Correspondence: xugy1990@163.com (G.X.); liuchong94@foxmail.com (C.L.)

**Abstract:** Effectively carrying out social responsibility is a critical strategy for the sustainable development of enterprises. Under the influence of institutional isomorphism, the relationship between corporate social responsibility and performance will be affected by the level of the peer firm's social responsibility and government subsidies. Based on institutional theory, this paper discusses the relationship between corporate social responsibility (CSR) and corporate performance, using relevant data from Chinese listed companies. The results show that there is an inverted U-shaped relationship between social responsibility and corporate performance; the peer firm's CSR and government subsidies weaken the inverted U-shaped relationship between CSR and corporate performance. The results provide useful theoretical insights for the performance of CSR.

**Keywords:** corporate social responsibility; government subsidies; peer firm's CSR

## 1. Introduction

Companies are now being expected to do more as part of their corporate social responsibility (CSR) activities, and these activities are becoming a corporate duty rather than a choice [1]. The reason behind this shift is that CSR can lead to excellent social standing, strengthen the competitiveness of the company and, thus, promote its sustainable development [2]. However, more investment in social responsibility is not always better; investments need to be kept at a certain level, otherwise CSR activities could put unbearable pressure on the business [3]. Not surprisingly, given the growing significance of these companies in the worldwide market, they have drawn a great deal of attention in recent studies [4–6]. There is a vociferous scholar who believes that social responsibility has a significant relationship with corporate performance, yet the body of empirical proof on this point is confusing. Supporters believe that social responsibility is the key to improving the legality and reputation of enterprises [7,8], while opponents argue that social responsibility will bring a large amount of financial pressure to enterprises, which is not conducive to the R&D and production of core products [9].

Results are inconsistent regarding the impact of CSR on firm-level outcomes [10]. Based on stakeholder theory and the resource-based view, some scholars have argued that CSR practices can reduce transaction costs by establishing a good reputation among stakeholders, improving the efficiency of business operations and, thus, promoting firm performance [11]. In contrast, trade-off theory suggests that core corporate resources should be used to maximize shareholder value rather than social responsibility [12]. Some scholars further suggested that excessive CSR practices will prevent companies from maximizing their resources and are detrimental to firm performance [9]. Additionally, a previous study also argued that the relationship between CSR and firm performance is uncertain because there are too many intervening factors [7,13]. Overall, these discoveries demonstrate that CSR shapes corporate-level outcomes in a myriad of ways.

Several causes led to the above hybrid findings. On the one hand, CSR and firm performance may be a nonlinear relationship because CSR has certain characteristics, such as being of a long-term nature [14,15] and being complex [16]. In addition, the relationship between CSR and firm performance may be influenced by certain essential boundary conditions, such as the political connections of enterprises [17] and economic policy uncertainty [18]. Surprisingly, we know relatively little about how industry-related factors impact the effectiveness of CSR. In fact, field cohesion is a crucial influencer of corporate motivation [19]. Therefore, companies tend to move their CSR levels closer to the industry average to align with industry institutional norms [2]. That is, peer firms' CSR may play an instrumental moderating position in the relationship of CSR and firm-level outcomes. Furthermore, the implementation of CSR practices cannot be separated from what is mandated in national policies, which is more obvious in developing countries [20]. Based on this, the objective of the current study is to address the nonlinear relationship between CSR and firm performance, along with the moderating roles of peer firms' CSR and government subsidy in this relationship.

The present study makes several contributions to the literature. First, this study shows that the relationship between CSR and firm performance is an inverted U-shape, hence, providing a theoretical reference for the level of CSR investment. Second, peer firms' CSR plays an important moderating role in the relationship between CSR and firm performance because of the "contagion effect". Third, the role of government subsidies is not all positive and needs to be controlled within a certain range. Following the literature synthesis, the data used, study methodology, results, and discussion are carried out. Study limitations and suggestions for future research are also explored.

## 2. Literature Review and Hypotheses Development

### 2.1. Corporate Social Responsibility

Firms should behave in ways that make them economically profitable or morally desirable, or both [21,22]. This means that firms should consider other activities outside of their day-to-day operations. With the increase in CSR research, some scholars have suggested that CSR helps improve social welfare, not just satisfy the economic or legal demands of the enterprise [23]. Thus, CSR can be defined as a commitment to enhance the well-being of society through the contributions of a firm's resources and discretionary business practices [22]. The concept of CSR has undergone a shift from compliance to value commitment [24], where the compliance view focuses on meeting government regulations, while the commitment perspective emphasizes making a beneficial contribution to society.

The reasons why companies would want to fulfill their social responsibilities mainly include ethical obligations, maintaining sustainability, improving legitimacy, and building reputation [25]. An ethical obligation here means that firms achieve business success in ethical ways, giving back resources to society without asking for a return [4]. The view of sustainability indicates that CSR can help companies generate direct economic benefits and lay a good foundation for their sustainable development [8]. Legitimacy emphasizes that CSR practices can alleviate the differences or conflicts between companies and stakeholders [26]. Finally, CSR has obvious altruistic tendencies, which can improve the company's reputation among customers, investors, and employees, while helping companies gain a reputation as an intangible asset [27].

The literature has suggested that CSR can be used as a strategic, competitive tool [28]. That is, social responsibility can be considered an anchor for corporate legalization and can provide institutional norms for the smooth entry of companies into other markets [29]. Moreover, as an innovative strategy, CSR practice can gain support from stakeholders and improve corporate competitiveness and performance [30]. However, CSR practices also have competitive characteristics and expendable attributes. Specifically, agency theory suggests that under the condition of limited corporate assets, CSR practices will inevitably occupy limited resources and create a crowding effect on the implementation of other corporate strategies, which will ultimately be detrimental to corporate performance im-

provement [31]. Meanwhile, it should be noted that a social responsibility strategy is easily imitated by other companies in the same industry, which weakens the competitive-enhancement effect of CSR. Therefore, the relationship between CSR and firm performance may be nonlinear.

### 2.2. Institutional Theory

Although institutional theory suggests that firms are embedded in and constrained by the institutional environment, they can also implement certain strategies to adapt to the institutional environment [32]. In short, institutional conditions may shape the propensity of firms to act in a socially responsible manner. For example, tax law is an overarching ownership regime that affects corporate charitable giving. Companies may also behave in a responsible manner for society if there are regulatory or cultural institutions that create the right incentives for this kind of behavior.

### 2.3. CSR and Firm Performance

CSR is often considered a poly-dimensional concept that encompasses economic, legal, ethical, and philanthropic aspects [21], and has been defined from several angles [33]. According to institutional theory [34], CSR is the strategic orientation of those firms that are capable of pursuing economic goals when implementing environmentally or socially desirable actions [34]. Institutional conditions are a significant factor in the implementation of CSR by companies [35]. In fact, this theory suggests that companies are more inclined to practice CSR if there are powerful and well-enforced national regulations, or if there is a structured and effective system of industry self-regulation to ensure such behavior [33,36].

CSR can help enterprises increase positive impacts and reduce negative external impacts [35,37]. This is in line with institutional theory, in which enterprises strive to improve legitimacy through a series of efforts. In this vein, the impact of social responsibility on corporate performance is mainly through the following two mechanisms: On the one hand, CSR affects the competitive mechanism of enterprises. The competition mechanism includes product competitiveness and enterprise competitiveness (legitimacy construction) [16,38]. On the other hand, social responsibility affects the consumption mechanism of enterprises. The consumption mechanisms mainly include cost consumption (funds, personnel, etc.) and imitation consumption (imitated by other enterprises) [39].

(1)    CSR affects the competitive mechanism of enterprises to improve performance.

Product competitiveness. CSR is a strategic behavior to effectively improve corporate performance as a differentiated product strategy [28]; it can have a beneficial impact on the construction of an enterprise's brand and customers' purchase behavior before later improving the competitiveness of enterprise products. In terms of brand building, CSR helps companies build a competitive advantage based on reputation [40]. The most important thing to note here is that CSR is a direct strategic behavior to increase brand influence and enhance the corporate image [27]. This is because with the concept of environmental protection gradually gaining popularity, consumers' desire products that minimize or eliminate any harmful impact on society and maximize the long-term beneficial effects on society [41]. By catering to this consumer psychology, corporate social responsibility raises customers' hopes for the company's products, thereby enhancing brand value and reputation. In terms of customer purchase behavior, CSR has a positive impact on customers' attitudes and behaviors [42]. Direct evidence in this area can be observed from products with higher prices. Consumers' purchase behavior is a conscious and well-thought-out choice based on individual moral beliefs. CSR can improve customer loyalty and satisfaction, allowing companies to have more pricing rights [43] and a higher enterprise valuation [44]. Therefore, in a competitive mechanism, CSR can effectively enhance corporate performance.

Enterprise competitiveness. Social responsibility can effectively enhance the legitimacy of enterprises and reduce enterprise risks, so as to improve the competitiveness of enterprises [35]. In terms of enhancing legitimacy, institutional theory suggests that CSR is part of a company's strategy to enhance legitimacy because institutions restrict behavior

among members of society and form the group rules that companies need to face. Only by following the guidance and constraints of the system can organizations or individuals obtain the legitimacy of operating in the new environment [45]. This is a pragmatic choice to avoid risks, magnify benefits, and create value for the company [46], because the effective implementation of social responsibility is a positive signal from the company, which makes the external stakeholders of the company optimistic about the future operation of the company, which will alleviate business pressure. At the same time, CSR can enhance the trust between internal and external stakeholders, establish social capital, and then reduce the transaction costs to maintain the stability of financial performance [47].

(2)　CSR will consume a lot and have a negative impact on the improvement of enterprise performance.

CSR will create a large amount of resource consumption, mainly because of two aspects: On the one hand, CSR may waste more valuable resources that should have helped shareholders maximize profits [12]. With the significant investment in CSR, it will gradually become an unbearable pressure for enterprises [39]. On the other hand, the product differentiation competition established by a CSR strategy can help enterprises establish well-known brands and a good corporate reputation. However, the competitiveness of enterprises based on this is relatively fragile because this mode of competition is easy to imitate and surpass. Therefore, the competitive advantage established by enterprises through social responsibility will converge with other enterprises, which will reduce their continuous competitiveness and lead to a decline in performance. The interaction effect between CSR and performance is shown in Figure 1. Therefore, we propose the following:

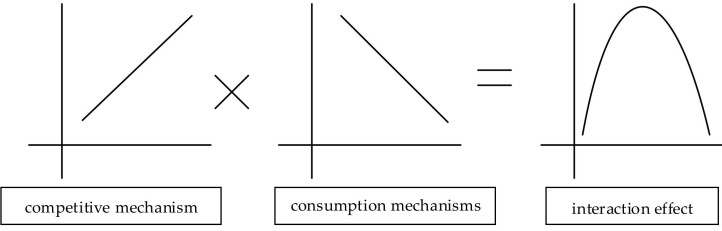

**Figure 1.** Interaction effect diagram of CSR and performance.

**Hypothesis 1.** *There is an inverted U-shaped relationship between CSR and firm performance.*

### 2.4. The Moderating Role of Peer Firms' CSR

This proved that everyone tends to do what others are doing, even if it is contrary to the information they receive [48]. If a certain behavior accounts for the vast majority of behaviors in the social network of enterprises, then social distancing [49] or collective punishment mechanisms [50] will have an enormous social impact on those enterprises or individuals who do not adopt this behavior, hence, compelling them to adopt this behavior. At the same time, since imitation behavior is typical in groups [51], the demonstration and normative effects of the industry can contribute to peer firm's CSR to a large extent and will influence the level of social responsibility in the focal firm.

First, the industry social responsibility level can affect the enterprise's competitive mechanism to exert an influence on enterprise performance. The level of a peer firm's CSR refers to the overall level of social responsibility within a certain industry, which represents the overall image of the industry and brings more influence than individual enterprises. An increase in the average level of social responsibility in the industry can weaken the competitive mechanism that CSR brings to the focal company [52]. This is mainly because, from the perspective of legitimacy in corporate competition, based on the normative and imitative influences in institutional theory [26], the level of corporate social responsibility will gradually converge with the level of social responsibility in the industry. This can

frustrate the original expectations of firms to improve their differentiated competition by fulfilling social responsibility, affecting the rapid improvement of corporate performance.

Second, as the level of peer firm's CSR increases, focal CSR can weaken the mechanism of CSR consumption. This is because, when the industry's level of social responsibility is high, focal companies can leverage the industry's reputation to improve their legitimacy. Companies can put more investment into corporate products and other areas that can help them gain performance. Therefore, we propose the following:

**Hypothesis 2.** *Peer firms' CSR weakens the inverted U-shaped relationship between CSR and firm performance.*

### 2.5. The Moderating Role of Government Subsidies

Government subsidies are an essential economic means for the government to intervene and regulate the micro behavior of enterprises, which can correct the distortion and externality of resource allocation under pure market regulation. As a voluntary social and moral commitment of corporations beyond legal obligations, CSR is not only beneficial to individuals and groups in society, but also a practical behavior, in line with the government's goal of creating public social value [53]. In this case, the government and the public expect firms to play a key role in economic development and make broader social contributions [8]. In the early literature on government subsidies, government subsidy initiatives consisted mainly of marketing assistance and financial programs [54]. Further, government subsidy initiatives can be divided into four subtypes: information-related, training-related, trade-flow-related, and financial-assistance-related subsidies [55].

(1)  Government subsidies will impact the competition mechanism of firms, thus, affecting firm performance.

The fulfillment of social responsibility is a kind of nonmarket strategic behavior conforming to government policy [56]; therefore, the state tends to reward firms for their social responsibility behaviors. However, some firms conduct rent-seeking behavior to obtain government subsidies [57], such as stating slogans but not implementing them, squeezing employees in disguise, and so forth, all of which affect the improvement of firm performance. At the same time, excessive government subsidies will cause firms to take unfair means of competition [39], because financial subsidies distort the competition norms of the market to a certain extent, which leads to unfair competition for other firms in the market. In addition, many studies have shown that excessive government subsidies will have a crowding-out effect on a firm's R&D expenditure, as government subsidies may lead to excessive dependence on government, which leads to inefficient operations and is not conducive to rapid improvements in firm performance [58].

(2)  Government subsidies affect the consumption mechanism of firms, which impacts firm performance.

Government subsidies can reduce the negative impact of consumption mechanisms by providing substantial help and support for the practice of CSR. At the same time, government subsidies play a role in signal transmission [55]. It not only provides relevant market information, planning, and training knowledge to the focal firm, but also provides positive investment signals to external investors [59], which helps to alleviate the financing constraints of the firm. In particular, it can help innovative companies relieve the pressure of product development and improve their product development ability [60], thus, weakening the negative impact of the social responsibility consumption mechanism and restraining the downward trend of performance. Therefore, we propose the following:

**Hypothesis 3.** *Government subsidies weaken the inverted U-shaped relationship between CSR and firm performance.*

### 3. Methods

The data used in this study are related to Chinese A-share listed companies from 2010–2020. China is currently the largest developing country. In a series of product safety and quality scandals in the last few years, Chinese society has become increasingly aware of the importance of CSR [2]. The current study utilizes four primary resources to collect data: the Accounting Research (CSMAR) database, company annual reports, the Hexun website, and Wind. These are widely used in studies of Chinese listed companies, and they provide relevant information on company background and financial statistics. We took Chinese listed companies on A-shares from 2010 to 2020 as samples to empirically test the relationship between CSR, peer firms' CSR, government subsidy, and firm performance. We did this because since 2010, China's CSR has been in the formal disclosure stage. At the same time, to ensure the reliability of the research results, we had to (1) eliminate the data samples of ST (special treatment firms), * ST (suspension from trading), and PT (Particular Transfer) enterprises [52]; (2) remove samples with seriously missing relevant values, and (3) exclude the insolvent sample. A total of 18,324 observations from 3137 enterprises were obtained. To prevent the influence of outliers on the regression results, we winsorized the main variables at the levels of 1% and 99%.

#### 3.1. Independent Variables

CSR is measured by the scores published on the Hexun website [61]. The evaluation system is investigated based on five items: investors, employees, suppliers, customers, environment, and community. The higher the score, the better the performance of CSR. The weights of these five aspects in the total score are 30%, 15%, 15%, 20%, and 20%, respectively.

#### 3.2. Dependent Variables

Currently, the measurement methods of enterprise performance mainly adopt ROA and ROE [62]. To better test the data, we used ROE as the measurement of enterprise performance and ROA as the alternative variable in the robustness test.

#### 3.3. Moderators

Peer firm's CSR refers to the average value of social responsibility of all enterprises in the industry [52]. We made a slight adjustment to eliminate the social responsibility score of the focus enterprise in the current year. This measurement is not simply used for averaging the CSR of the same industry; its advantage is that it can represent a cross-network relationship. The level of industrial social responsibility of different enterprises in the same industry in the same year is also different. It does not only express the impact of other enterprises in the same industry on the focus of CSR, but the enterprises can also participate in the same industry groups to affect other individuals. Government subsidies represent the degree of government support for enterprises, which is calculated as the total amount of government subsidies actually obtained by the enterprises.

#### 3.4. Control Variable

Several control variables were included: Firm age represents the number of years since the company was founded. Firm size has been shown to affect CSR ratings, so we included firm size as a control, which is measured as the natural log of total assets [39]. For firm nature, there are significant differences between state-owned enterprises and non-state-owned enterprises when it comes to resource advantages, values, and strategic objectives. State-owned enterprises not only have higher legitimacy, but also enjoy preferential policies, such as government finance [63]. Equity concentration is the sum of the shareholding ratios of the top 10 major shareholders of the company and affects corporate decision making and investment in CSR. Slack resources can affect the extent to which companies are willing and able to invest in social responsibility initiatives [11]. We have measured slack resource as the ratio of current assets to current liabilities. The relevant explanations of the variables are shown in Table 1.

**Table 1.** Explanation of variables.

| Indicators | Variable Code | Definitions |
|---|---|---|
| Independent variable | | |
| Enterprise performance | roe | Ratio of net income to net assets |
| Dependent variable | | |
| CSR | CSR | The actual score of each company in Hexun |
| Moderator variables | | |
| Peer firms' CSR | Psrl | Average social responsibility score of all companies in the industry in which the focal company is located in a given year, excluding the focal company |
| Government Subsidy | sub | The total amount of government subsidies actually obtained by enterprises |
| Control variables | | |
| Firm Age | age | the number of years since the company was founded |
| Firm Size | lnsize | Natural logarithm of total corporate assets |
| Firm Nature | state | The value is 1 when the enterprise is a state-owned enterprise, otherwise it is 0 |
| Ownership Concentration | shrcr | Percentage of shares owned by the top 10 shareholders |

## 4. Analyses

### 4.1. Analysis and Results

The outcomes regarding descriptive statistics and correlation analysis are shown in Table 2. The average score of CSR is 26.50, with a standard deviation of 17.03, suggesting that the level of CSR in China needs to be improved (the total score is 100); the CSR level is quite different in the different enterprises. Furthermore, Table 3 shows that the correlation coefficients are all lower than 0.5, suggesting that there are no critical issues. We further opted to use the variance inflation factor (VIF), VIF = 1.42, and the unreported results indicate no serious multicollinearity problem. In the concurrent phase, to avoid the influence of heteroscedasticity, we obtained a $p$-value of 0.000 using the white test, which significantly rejected the original hypothesis and proved the existence of heteroscedasticity. To avoid the influence of heteroscedasticity on the regression model, weighted least squares (WLS) analysis was used to estimate model parameters. As a first step, we estimated a linear regression model using ordinary least squares (OLS). The estimated value of the residual term was obtained as $\hat{u}_i$, which led to $\ln\hat{u}_i^2$. All independent variables were regressed with $\ln\hat{u}_i^2$, and then the fitting value of the explained variable was obtained. We calculated the weight $\hat{h}_i$ and took $\frac{1}{\hat{h}_i}$ as the weight, using WLS to handle the regression model.

**Table 2.** Descriptive statistics and Pearson correlations.

| Variable | Mean | SD | 1 | 2 | 3 | 4 | 5 | 6 | 7 | 8 | 9 |
|---|---|---|---|---|---|---|---|---|---|---|---|
| 1. roe | 6.977 | 9.695 | 1 | | | | | | | | |
| 2. state | 0.397 | 0.489 | −0.067 *** | 1 | | | | | | | |
| 3. lnsize | 22.08 | 1.288 | 0.077 *** | 0.391 *** | 1 | | | | | | |
| 4. lnage | 1.946 | 0.943 | −0.108 *** | 0.470 *** | 0.436 *** | 1 | | | | | |
| 5. Shrcr | 59.67 | 15.33 | 0.199 *** | −0.100 *** | 0.088 *** | −0.442 *** | 1 | | | | |
| 6. slack | 2.903 | 4.946 | 0.038 *** | −0.190 *** | −0.264 *** | −0.292 *** | 0.122 *** | 1 | | | |
| 7. CSR | 26.50 | 17.03 | 0.398 *** | 0.140 *** | 0.294 *** | 0.055 *** | 0.122 *** | −0.001 | 1 | | |
| 8. Psrl | 26.09 | 5.987 | 0.087 *** | 0.156 *** | 0.092 *** | 0.080 *** | 0.024 *** | 0.005 | 0.266 *** | 1 | |
| 9. sub | 15.81 | 1.898 | 0.076 *** | 0.173 *** | 0.417 *** | 0.095 *** | 0.056 *** | −0.103 *** | 0.217 *** | 0.125 *** | 1 |

Note: *** indicate significance at the 1% level.

First, we analyzed the regression model between CSR and performance. M1 contains only the control variables and M2 increases CSR and CSR2. The results show that the CSR coefficient (β = 0.011, $p < 0.001$) is significantly positive, and the coefficient of CSR2 (β = −0.012, $p < 0.001$) is significantly negative, which preliminarily proves that there is an inverted U-shaped relationship between social responsibility and internationalization performance. To make the results more accurate, we used the U-test command in Stata to further verify. The results show that the extreme value of CSR is 45.93407, which is just within the 95% fielder range (45.476864; 46.408596). Further, $p = 0.0000 < 0.01$ for the overall

test of the inverted U-shaped relationship, so there is an inverted U-shaped relationship between the independent and dependent variables and Hypothesis 1 holds.

**Table 3.** Regression analysis.

| Variables | M1 | M2 | M3 | M4 | M5 |
|---|---|---|---|---|---|
| CSR | | 1.083 *** | 1.583 *** | 1.496 *** | 1.887 *** |
| | | (0.011) | (0.047) | (0.088) | (0.095) |
| $CSR^2$ | | −0.012 *** | −0.019 *** | −0.020 *** | −0.026 *** |
| | | (0.000) | (0.001) | (0.001) | (0.001) |
| Psrl | | | 0.266 *** | | 0.025 *** |
| | | | (0.028) | | (0.028) |
| CSR × Psrl | | | 0.020 *** | | −0.019 *** |
| | | | (0.002) | | (0.002) |
| $CSR^2$ × Psrl | | | 0.0003 *** | | 0.0002 *** |
| | | | (0.000) | | (0.000) |
| sub | | | | 0.401 *** | 0.331 *** |
| | | | | (0.084) | (0.085) |
| CSR × sub | | | | −0.025 *** | −0.019 * |
| | | | | (0.006) | (0.006) |
| $CSR^2$ × sub | | | | 0.001 *** | 0.0004 ** |
| | | | | (0.000) | (0.0001) |
| State | −0.845 *** | −0.763 *** | −0.767 *** | −0.800 *** | −0.787 *** |
| | (0.146) | (0.116) | (0.116) | (0.116) | (0.116) |
| lnsize | 0.807 *** | 0.140 ** | 0.149 ** | 0.021 | −0.021 *** |
| | (0.058) | (0.047) | (0.047) | (0.050) | (0.050) |
| lnage | −0.550 *** | 0.464 *** | 0.043 *** | 0.527 *** | 0.560 *** |
| | (0.079) | (0.063) | (0.004) | (0.064) | (0.064) |
| shrcr | 0.087 *** | 0.042 *** | 0.043 | 0.043 *** | 0.044 |
| | (0.005) | (0.004) | (0.004) | (0.004) | (0.004) |
| Slack | −0.020 *** | −0.049 *** | −0.046 *** | −0.047 *** | −0.045 *** |
| | (0.006) | (0.004) | (0.004) | (0.004) | (0.004) |
| N | 18,324 | 18,324 | 18,324 | 18,324 | 18,324 |
| $R^2$ | 0.0535 | 0.4116 | 0.4155 | 0.4144 | 0.4180 |
| Adj $R^2$ | 0.0533 | 0.4114 | 0.4151 | 0.4141 | 0.4176 |
| F | 207.27 | 1830.40 | 1301.55 | 1295.88 | 1011.78 |

Note: This table reports the regression results. Robust standard errors in parentheses. See Table 1 for variables definitions. ***, **, and * indicate significance at the 1%, 5%, and 10% levels.

To test this moderating effect, we drew on the model of Haans et al. (2016) [64]:

$$y = \beta_0 + \beta_1 x + \beta_2 x^2 + \beta_3 xz + \beta_4 x^2 z + \beta_5 z$$

where y is the dependent variable, X is the independent variable, and Z is the regulating variable. In this model, we aimed to observe whether the quadratic coefficient of the independent variable ($\beta_2$) and the coefficient of the interaction between the independent variable and the regulating variable ($\beta_4$) are significant, hence, allowing us to judge the regulation direction of the regulation effect according to the symbols $\beta_2$ and $\beta_4$. In model 3, we could test the moderating effect of a peer firm's CSR. The results show that $\beta_2 = -0.019$ ($p < 0.001$) and $\beta_4 = 0.0003$ ($p < 0.01$) are significant, and the symbols are the opposite. Here, the level of a peer firm's CSR eases the inverted U-shaped radian. Thus, Hypothesis 2 is supported (Table 2).

In model 4, the role of a government subsidy in moderating the relationship between CSR and performance was examined in our moderate mediation analysis. The quadratic coefficient $\beta2 = -0.020$ ($p < 0.001$) and quadratic and interaction terms $\beta_4 = 0.001$ ($p < 0.001$) are both negative and significant. This also eases the inverted U-shaped radian. This result supports Hypothesis 3. In model 5, we added all variables to the regression model. The results show that the main and regulatory effects are significant.

### 4.2. Robustness Checks

To further troubleshoot other explanations and to confirm our main findings, we performed some robustness tests. First, we replaced the dependent variable to further verify the relevant empirical results. We replaced the dependent variable ROE (Rate of Return on Common Stockholders' Equity) with ROA (Return on Assets), and the test results are consistent with the previous ones. We increased the control variable because CSR needs significant funds for investment and because when the enterprise has more liabilities or less cash flow, this may inhibit the investment of CSR. Therefore, to further verify the accuracy of the results, we added two control variables: corporate liabilities and cash flow. The results are consistent with previous studies. Third, the endogenous problem is solved. There may be an endogenous problem of mutual causality between CSR and performance; that is, better corporate performance makes enterprises have good social responsibility. To solve this problem, the data with a one period lag of explanatory variables were mainly used in the regression estimation, making the research conclusions more reliable and robust.

## 5. Conclusions and Discussion

Prior studies on CSR and firms' performance have presented mixed findings [65]. In the current paper, we have shown that the impact of CSR on corporate performance is nonlinear. There is an inverted U-shaped relationship between CSR and firm performance. This paper examines the reasons for the mixed results from a long-term perspective. Most previous studies examined the relationship between CSR and firms' performance from a short-term perspective [66]. Prior research has also ignored the cost pressures and the risk of imitation associated with long-term socially responsible practices [56]. The findings of this paper remedy the shortcomings of previous studies and provide a new empirical test of the relationship between CSR and performance.

Peer firms' CSR will also play a prominent role, affecting the relationship between CSR and a firm's performance. The findings of this paper provide further evidence that the peer firm's CSR can have a significant impact on the focal company [66,67]. Institutional pressure has led to an industry convergence effect on corporate socially responsible behavior [68]. Peer firms' CSR enhances the overall industry image, on the one hand, and gives companies the energy to make efforts to improve their core competencies, such as products and sales, on the other hand. Therefore, peer firms' CSR weakens the inverted U-shaped relationship between CSR and firm performance.

However, higher government subsidies are not always better, so it is necessary to control for the positive impact on the performance of the company within a reasonable range. The results of this study enrich and refine the shortcomings of previous studies that focused only on the positive impact of government subsidies [69], although government subsidies can increase the legitimacy of enterprises and reduce their consumption. However, government subsidies have the potential to lead to rent-seeking behavior by firms to obtain subsidies, for example, by chanting slogans without practicing them [57]. Therefore, Government subsidies weaken the inverted U-shaped relationship between CSR and firm performance.

### 5.1. Theoretical Implications

The current paper makes some significant theoretical contributions to the literature. First, the present paper has emphasized both moral attributes and competitive attributes. These two factors have rarely been considered together from a holistic perspective. Most previous studies emphasized the moral attribute of social responsibility [33], ignoring its competitive attribute. Recently, a small number of studies have begun to focus on the strategic competitive attribute of social responsibility [70,71] but have still ignored the consumption attribute, especially "being imitated", which makes the previous research on social responsibility incomplete. In the current paper, we have shown that CSR affects performance through competition mechanisms and consumption mechanisms. On the

one hand, CSR also provides differentiated competitiveness for products and improves the legal status of these products [33]. On the other hand, with an increase in enterprise investment, this differentiated competitive strategy is gradually imitated by other enterprises. The disappearance of this strategic advantage weakens the positive impact of CSR on performance. This shows that an investment in CSR should be reasonably allocated based on the difference between the company's own ability and its strategic objectives. This conclusion also further explains the reasons for the mixed empirical results between CSR and performance, providing a theoretical reference for the follow-up research on the relationship between CSR and performance.

Second, the current study demonstrates the usefulness of institutional theory in explaining the adoption of CSR strategies by firms. Corporate behavior in the same industry will have an enormous impact on corporate behavior [72]. The institutional pressure of peers makes it difficult for enterprises to deviate from the norms in their industry [73]. Therefore, a peer firm's CSR will have a meaningful impact on the relationship between CSR and performance. However, previous studies have mostly ignored this critical influencing factor [66]. Through empirical testing, the current paper indicates the importance of peer firms' CSR.

Finally, as an essential supplement to the "invisible hand" of the market, government subsidies can improve resource allocation and enterprise externality under market regulation [74]. However, most previous studies have described the positive impact of government subsidies, ignoring their negative impact [75]. In particular, the large-scale direct subsidy policy may have an adverse impact on product research and development, promote the behavior of market subjects to change in the direction of increasing their own interests because of the subsidy policy, and cause a decreased efficiency problem [58]. This may cause enterprises to falsely increase their social responsibility investment to get government subsidies, which will bring more tremendous financial pressure to enterprises. Based on this, a general insight from our study is that government subsidies are complex and need to be set within a reasonable range, which can help provide theoretical support for the rational planning of government subsidies.

*5.2. Practical Implications*

First, the impact of CSR on performance should not be viewed only in terms of the company itself (including legitimacy building, cost loss, etc.), but also by considering the degree of imitation by competitors. Once other enterprises imitate the focal firm, the competitive advantage brought about by their own social responsibility strategy will decline or even disappear. Therefore, enterprises should strive to carry out differentiation strategies, such as product R&D, to maintain their long-term competitiveness. Second, more social responsibility investment is not always the best solution. The influence of social responsibility should not be a value judgment of "black or white" but should instead balance the enterprise's own ability and overall strategy, allowing it to "act according to its ability", based on the enterprise's resource base and strategic objectives. Third, the social responsibility level of the same industry is the reference of corporate social responsibility. Once the social responsibility level of the industry is low, the focus should be to take the performance of social responsibility as a differentiated competitive strategy to enhance its influence. Policymakers should also actively guide the improvement of the social responsibility level of the whole industry. When the social responsibility level of the industry is high, enterprises can rely on the high reputation of the industry and strive to improve their core competitiveness, such as product R&D, channel construction, and so forth. Finally, the government needs to set up reasonable subsidy standards, according to the development of the industry and market. Excessive government subsidies will lead to rent-seeking behavior or crowding-out effects.

*5.3. Limitations and Future Research*

Based on the research content and development of practice, there are still two points that need further research. First, the current research usually regards social responsibility as a whole. However, an enterprise may exhibit both responsible and irresponsible behavior (Price and Sun, 2017) [65]. Future research should distinguish and integrate responsibility and irresponsibility, exploring their impact on enterprise performance. Second, only the data of listed companies in China will cause a certain sample selectivity deviation, so the samples of non-listed companies should be gradually added in future research. Third, currently, government subsidies are mostly divided into R&D subsidies and non-R&D subsidies. Different kinds of government subsidies will have different effects on enterprises. Future research should distinguish among the different effects of various subsidies.

**Author Contributions:** Conceptualization, W.Z. and G.Y.; methodology, G.X.; software, C.L.; validation, D.D., M.H. and W.Z.; formal analysis, W.Z.; investigation, G.X.; resources, G.Y.; data curation, G.Y.; writing—original draft preparation, G.X.; writing—review and editing, W.Z. and C.L.; visualization, D.D.; supervision, M.H.; project administration, G.X. and C.L.; funding acquisition, W.Z. All authors have read and agreed to the published version of the manuscript.

**Funding:** Key Projects of Philosophy and Social Sciences Research of Ministry of Education of China, 17JZD020.

**Institutional Review Board Statement:** Not applicable.

**Informed Consent Statement:** Not applicable.

**Data Availability Statement:** The relevant data for the current study comes from the CSMAR database and Hexun.com. (accessed on 4 April 2021).

**Conflicts of Interest:** None of the authors have any conflict of interest to report.

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
