# Peer review of "CSR and Long-Term Corporate Performance: The Moderating Effects of Government Subsidies and Peer Firm’s CSR"

_sustainability, doi:10.3390/su14095543_

Round 1

Reviewer 1 Report

The study shows a good scientific structure, valid research design and meaningful results and conclusions. There aren’t also major concerns about plagiarism (Turnitin reports a 16% similarity with other sources, two of them with a >2%). Anyway, I recommend a major revision to solve some deficiencies:

  • Please clarify what abbreviations “ST, * ST, and PT enterprises” stand for.
  • Please clarify this sentence, since I do not understand the meaning of “sample” in it… I guess that “observations” would be more appropriate: “A total of 18,324 samples from 3,137 enterprises were obtained”.
  • English writing should be revised by a copy-editing service.
  • In the method section a subsection should be included explaining the method of analysis. Why not panel data analysis?
  • I suggest adding a discussion section where results are discussed with relation to previous literature.

The literature review should include a bigger explanation of the vast previous literature dealing with the analysis of the relationship between CSR and corporate (financial) performance. Moreover, there are no references from the target journal. Please add this reference (https://doi.org/10.1016/j.jik.2021.11.001) in the introduction when explaining that “more investment in social responsibility is not always better; investments need to be kept within a certain level; otherwise, CSR activities could put 25 unbearable pressure on the business”, saying that CSR investments should, thus, be economically justified and that valid economic assessment methods need to be applied. Also cite this work in the literature review about previous studies dealing with the CSR-financial performance relation (https://doi.org/10.3390/su11041211). 

Author Response

Responses to Reviewer #1:

Thank you for offering constructive suggestions for improving the manuscript. We have revised the paper following your suggestions. All changes are highlighted in yellow in the revision. Here are our responses to your comments:

1Please clarify what abbreviations “ST, * ST, and PT enterprises” stand for.

Response:

We really appreciate for your constructive comment. As suggested, we have included the full names of these abbreviations in the article, as described on page 6 of the article.

Change the content to “ST (special treatment firms), * ST (suspension from trading), and PT(Particular Transfer)enterprises.”

Note: ST, * ST, and PT are labels given to some of the stocks in the Chinese stock market that are in trouble.

ST(special treatment):When a company loses money for two consecutive years or its net assets are below the par value of the stock, "ST" will be added before the stock name, which means "special treatment".

If, in the third year, the company's operation does not improve and remains in a loss position, the stock name will be preceded by "*" in addition to "ST", which means "*ST", meaning that there is a risk of delisting.

"PT"(Particular Transfer). This is a "special transfer service" designed to provide a circulation channel for suspended stocks. Stocks subject to such "special transfer" are referred to as "PT shares" on the Shanghai and Shenzhen stock exchanges by prefixing the abbreviation "PT" with the abbreviation "PT".

2Please clarify this sentence, since I do not understand the meaning of “sample” in it… I guess that “observations” would be more appropriate: “A total of 18,324 samples from 3,137 enterprises were obtained”.

Response:

Thank you very much for the reminder. In the Stata statistics software, we counted a total of 18,324 observations. so, we should change "sample" to "observations". See yellow section on page 6 for specific changes

3English writing should be revised by a copy-editing service.

Response:

In order to better present our research ideas, we submitted the article to a professional organization for English language retouching to revise the grammar and expression of this paper.

4In the method section a subsection should be included explaining the method of analysis. Why not panel data analysis?

Response:

Your advice is very instructive. Thank you very much for your professional advice. However, the main considerations for the OLS approach in this paper are as follows: first, we performed a heteroskedasticity test before conducting the regression analysis. The results of the study showed that the regression model had serious heteroskedasticity problems. In order to deal with this problem more rigorously, we adopt the WLS method to deal with the heteroskedasticity problem. This solution matches better with the OLS regression model.

Second, the OLS approach is the most common and basic estimation method in regression models. Numerous scholars have used this method for regression analysis. For example:Chen et al.,2021、Crilly et al.,2016、Doh et al.,2010.

1、Chen S, Chen Y, Jebran K. Trust and corporate social responsibility: From expected utility and social normative perspective[J]. Journal of Business Research, 2021,134:518-530.

2、Crilly D, Ni N, Jiang Y. Do-no-harm versus do-good social responsibility: Attributional thinking and the liability of foreignness[J]. Strategic Management Journal, 2016,37(7):1316-1329.

3、Doh J P, Howton S D, Howton S W, et al. Does the Market Respond to an Endorsement of Social Responsibility? The Role of Institutions, Information, and Legitimacy[J]. Journal of Management, 2010,36(6):1461-1485.

5I suggest adding a discussion section where results are discussed with relation to previous literature. The literature review should include a bigger explanation of the vast previous literature dealing with the analysis of the relationship between CSR and corporate (financial) performance. Moreover, there are no references from the target journal. Please add this reference (https://doi.org/10.1016/j.jik.2021.11.001) in the introduction when explaining that “more investment in social responsibility is not always better; investments need to be kept within a certain level; otherwise, CSR activities could put 25 unbearable pressure on the business”, saying that CSR investments should, thus, be economically justified and that valid economic assessment methods need to be applied. Also cite this work in the literature review about previous studies dealing with the CSR-financial performance relation (https://doi.org/10.3390/su11041211).

Response:

We really appreciate for your constructive comment. At your suggestion, we have added the references you suggested to the article. Such additions make our article more complete. The details are in the third and tenth references on the first page.

We have also added a discussion section to the article based on your suggestions. The details are as follows:

Conclusion and discussion

Prior studies on CSR and firm’s performance have presented mixed findings[65]. In the current paper, we have shown that the impact of CSR on corporate performance is nonlinear. There is an inverted U-shaped relationship between CSR and firm performance. This paper examines the reasons for the mixed results from a long-term perspective. Most previous studies have examined the relationship between CSR and firm’s performance from a short-term perspective[66]. And, prior research has ignored the cost pressures and the risk of imitation associated with long-term socially responsible practices[56]. The findings of this paper remedy the shortcomings of previous studies and provide a new empirical test of the relationship between CSR and performance.

Peer firm’s CSR will also play a prominent role, affecting the relationship between CSR and firm’s performance. The findings of this paper provide further evidence that the peer firm’s CSR can have a significant impact on the focal company[66,67]. Institutional pressure has led to an industry convergence effect on corporate socially responsible behavior[68]. Peer firm’s CSR enhances the overall industry image on the hand, and gives companies the energy to make efforts to improve their core competencies such as products and sales on the other hand. So, Peer firms’ CSR weakens the inverted U-shaped relationship between CSR and firm performance.

However, government subsidies are not higher, the better, so it is necessary to control for the positive impact on the performance of the company within a reasonable range. The results of this study enrich and refine the shortcomings of previous studies that focused only on the positive impact of government subsidies[69]. Although government subsidies can increase the legitimacy of enterprises and reduce their consumption. However, government subsidies have the potential to lead to rent-seeking behavior by firms to obtain subsidies, for example, by chanting slogans without practicing[57]. So, Government subsidies weaken the inverted U-shaped relationship between CSR and firm performance.

Reviewer 2 Report

.

Author Response

Thank you very much for your support and affirmation of this article

Reviewer 3 Report

The article is interesting, and the researched problem has scientific potential. However, some problems need to be solved:

  1. Literature review should include more recent sources (2019-2022) and must be enriched with relevant references.

 E.g. :

Hu, Q.; Zhu, T.; Lin, C.-L.; Chen, T.; Chin, T. Corporate Social Responsibility and Firm Performance in China’s Manufacturing: A Global Perspective of Business Models. Sustainability 2021, 13, 2388. https://doi.org/10.3390/su13042388

Nicolescu, M. M., & Vărzaru, A. A. (2020). Ethics and disclosure of accounting, financial and social information within listed companies. Evidence from the Bucharest Stock Exchange. New Trends in Sustainable Business and Consumption, 73.

Ying, M.; Tikuye, G.A.; Shan, H. Impacts of Firm Performance on Corporate Social Responsibility Practices: The Mediation Role of Corporate Governance in Ethiopia Corporate Business. Sustainability 2021, 13, 9717. https://doi.org/10.3390/su13179717

  1. Data processing is performed using descriptive statistics, correlation analysis and regression. The article would gain value if SEM were used to establish the relationships among variables.
  1. In my opinion, a section of discussion should be built in the context of dialogue with researchers in the literature review.

The article presents scientific value and can be published after a careful review of the reported issues.

Author Response

Responses to Reviewer #3:

1Literature review should include more recent sources (2019-2022) and must be enriched with relevant references.

Response:

This is a wonderful comment you made. At your suggestion, we have reorganized the latest literature and added it to the article. This made the article more fleshed out and meaningful. For example,we have added the following references:

1.Ying, M.; Tikuye, G.A.; Shan, H. Impacts of Firm Performance on Corporate Social Responsibility Practices: The Mediation Role of Corporate Governance in Ethiopia Corporate Business. Sustainability-Basel 2021, 13, 9717.

2.Okafor, A.; Adeleye, B.N.; Adusei, M. Corporate social responsibility and financial performance: Evidence from U.S tech firms. J Clean Prod 2021, 292, 126078.

3.Hu, Q.; Zhu, T.; Lin, C.; Chen, T.; Chin, T. Corporate Social Responsibility and Firm Performance in China’s Manufacturing: A Global Perspective of Business Models. Sustainability-Basel 2021, 13, 2388.

2Data processing is performed using descriptive statistics, correlation analysis and regression. The article would gain value if SEM were used to establish the relationships among variables.

Response:

Thank you very much for your constructive comments. Structural equation modeling is a statistical method to analyze the relationship between variables based on the covariance matrix of the variables and is an important tool for multivariate data analysis. The SEM approach has been working very successfully in the data analysis of questionnaires. However, since this paper uses secondary data to find proxies for key variables, OLS regression analysis is more appropriate. In future studies, when we include variables such as personal social responsibility and take a questionnaire to obtain data, we will use structural equation modeling.

3In my opinion, a section of discussion should be built in the context of dialogue with researchers in the literature review.

Response:

Your suggestion is very beneficial to the sublimation of the research contribution of this paper. In order to dialogue with the previous research literature, the following points have been added to this paper. The details are as follows:

5 Conclusion and discussion

Prior studies on CSR and firm’s performance have presented mixed findings[65]. In the current paper, we have shown that the impact of CSR on corporate performance is nonlinear. There is an inverted U-shaped relationship between CSR and firm performance. This paper examines the reasons for the mixed results from a long-term perspective. Most previous studies have examined the relationship between CSR and firm’s performance from a short-term perspective[66]. And, prior research has ignored the cost pressures and the risk of imitation associated with long-term socially responsible practices[56]. The findings of this paper remedy the shortcomings of previous studies and provide a new empirical test of the relationship between CSR and performance.

Peer firm’s CSR will also play a prominent role, affecting the relationship between CSR and firm’s performance. The findings of this paper provide further evidence that the peer firm’s CSR can have a significant impact on the focal company[66,67]. Institutional pressure has led to an industry convergence effect on corporate socially responsible behavior[68]. Peer firm’s CSR enhances the overall industry image on the hand, and gives companies the energy to make efforts to improve their core competencies such as products and sales on the other hand. So, Peer firms’ CSR weakens the inverted U-shaped relationship between CSR and firm performance.

However, government subsidies are not higher, the better, so it is necessary to control for the positive impact on the performance of the company within a reasonable range. The results of this study enrich and refine the shortcomings of previous studies that focused only on the positive impact of government subsidies[69]. Although government subsidies can increase the legitimacy of enterprises and reduce their consumption. However, government subsidies have the potential to lead to rent-seeking behavior by firms to obtain subsidies, for example, by chanting slogans without practicing[57]. So, Government subsidies weaken the inverted U-shaped relationship between CSR and firm performance.

Also, in the theoretical contributions section, we have added some references to respond to the previous studies. For example,

In the current paper, we show that CSR affects performance through competition mechanisms and consumption mechanisms. On the one hand, CSR also provides differentiated competitiveness for products and improves the legal status of these products[33]. On the other hand, with an increase of enterprise investment, this differentiated competitive strategy is gradually imitated by other enterprises.

Finally, as an essential supplement to the “invisible hand” of the market, government subsidies can improve resource allocation and enterprise externality under market regulation[74]. However, most previous studies have described the positive impact of government subsidies, ignoring their negative impact[75].

Round 2

Reviewer 1 Report

I acknowledge the efforts of the authors to consider the suggestions made in the first review report. The scientific quality of the article has been improved. As a minor matter, I suggest reconsidering where the discussion of results stands. If the content does not have enough entity to constitute an individual section, perhaps better than including it in the conclusions section would be to include it in the results section.

Reviewer 3 Report

The paper can be publised in current form.